# SWAAT Bioinformatics Workflow for Protein Structure-Based Annotation of ADME Gene Variants

**DOI:** 10.3390/jpm12020263

**Published:** 2022-02-11

**Authors:** Houcemeddine Othman, Sherlyn Jemimah, Jorge Emanuel Batista da Rocha

**Affiliations:** 1Sydney Brenner Institute for Molecular Bioscience, Faculty of Health Sciences, University of the Witwatersrand, 9 jubilee Road, Parktown, Johannesburg 2193, South Africa; jdarocha1@gmail.com; 2Department of Biotechnology, Bhupat and Jyoti Mehta School of Biosciences, Indian Institute of Technology Madras, Chennai 600036, India; sherlyn.jemimah@hotmail.com; 3National Health Laboratory Service, Division of Human Genetics, Faculty of Health Sciences, School of Pathology, University of the Witwatersrand, Johannesburg 2001, South Africa

**Keywords:** variant effect prediction, pharmacogenomics, energy, entropy, ADME genes, Nextflow

## Abstract

Recent genomic studies have revealed the critical impact of genetic diversity within small population groups in determining the way individuals respond to drugs. One of the biggest challenges is to accurately predict the effect of single nucleotide variants and to get the relevant information that allows for a better functional interpretation of genetic data. Different conformational scenarios upon the changing in amino acid sequences of pharmacologically important proteins might impact their stability and plasticity, which in turn might alter the interaction with the drug. Current sequence-based annotation methods have limited power to access this type of information. Motivated by these calls, we have developed the Structural Workflow for Annotating ADME Targets (SWAAT) that allows for the prediction of the variant effect based on structural properties. SWAAT annotates a panel of 36 ADME genes including 22 out of the 23 clinically important members identified by the PharmVar consortium. The workflow consists of a set of Python codes of which the execution is managed within Nextflow to annotate coding variants based on 37 criteria. SWAAT also includes an auxiliary workflow allowing a versatile use for genes other than ADME members. Our tool also includes a machine learning random forest binary classifier that showed an accuracy of 73%. Moreover, SWAAT outperformed six commonly used sequence-based variant prediction tools (PROVEAN, SIFT, PolyPhen-2, CADD, MetaSVM, and FATHMM) in terms of sensitivity and has comparable specificity. SWAAT is available as an open-source tool.

## 1. Introduction

Absorption, Distribution, Metabolism, and Excretion (ADME) genes are key players determining the pharmacokinetic properties of a drug. More than 300 genes have been identified to contribute to the ADME properties. The function of 32 core ADME genes has been confirmed by extensive studies [1]. Drug response depends on the variants of the ADME genes found in an individual. For instance, depending on the genetic variability, the well-studied *CYP2D6* ADME gene is associated with ultrarapid, intermediate, or poor metabolization of tamoxifen [2]. The polymorphism of ADME genes is a distinctive property of different populations [3,4,5,6]. For example, the rs1050828 T variant is known to cause an acute deficiency in *G6PD*. Its occurrence among genetically diverse groups (reflected in self-identify) in South Africa has been shown to vary significantly even among geographically neighboring populations [3]. Moreover, clinically actionable variants of ADME genes deviate significantly between Sub-Saharan Africa and other worldwide populations [7].

The characterization of common variants in ADME genes was the main focus of pharmacogenetic studies, which allowed for setting up clinical directives in general populations. However, these variants cannot solely explain the wide range of observed drug-response phenotypes [8]. In particular, rare variants were shown to have an important share of the genetic makeup and were identified for their putative functional implication in ADME genes [5,7,9,10]. The important status of rare variants has been also highlighted recently as critical to consider in deriving personalized medicine strategies [11]. Sequencing costs are expected to drop significantly in the upcoming years [12], which will improve our insight into the genetic variability at high-resolution levels. More data from ethnically diverse groups will become available, which necessitates reliable computational tools for variant interpretation and prioritization.

Variant functional prediction tools integrate a variety of features including conservation, amino acid properties, domain mapping, and annotation data [13]. Other algorithms include machine learning prediction tools trained over a large set of variants with known clinical and functional outcomes [14]. Variant effect predictors currently cover a plethora of tools and workflows: SIFT, PolyPhen, CADD, and Ensembl Variant Effect Predictor (VEP) are widely used in practice. Despite the significant improvement, variant functional prediction tools are constrained by many challenges including the lack of sensitivity [15], lack of proper training datasets [16], inappropriate benchmarking [13], and inflated accuracy reporting [14]. In the context of ADME genes, variant functional prediction tools are limited to sequence-based analysis, which might not capture scenarios such as conformational perturbation, drug–protein interactions, and others. As a subject for structural bioinformatics, these types of scenarios play an important role in explaining the heterogeneity of phenotypes related to precision medicine [17,18]. The integration of structural effects in genome analysis workflows is hindered by the inefficiency and the lack of reliable protein structures. Nevertheless, several tools were developed to predict the effect of DNA missense mutations/variants on protein structures [19,20,21,22]. However, these tools are not designed to accommodate the analysis of multiple variants, and they are usually employed as supplemental tools used at late stages of the downstream analysis process.

The ability to compute and retrieve attributes of genetic variants is an important step for characterizing their functional impact. This adds more complexity in the interpretation of the biological and clinical meaning, which is out of the reach of manual processing. In the last decade, there has been a significant advance in machine learning approaches applied in variant discovery and the functional prediction of missense variants [23,24]. These methods can integrate a large number of features, extract information relevant to the impact on the function and predict with high accuracy the outcome of the genetic variability. Moreover, the application of structural analysis has become an operating terrain for machine learning methods thanks in part to the availability of reliable datasets that offer measures of the direct thermodynamic impact on protein structures. Since demand is on the rise to understand, at the atomic scale, the mechanisms underlying protein mutations, the exploitation of machine learning approaches brings compelling promises.

This paper presents the Structural Workflow for the Annotation of ADME Targets (SWAAT) as a novel computational tool that allows studying the structural implication of missense variants. We explain the extent of the application of SWAAT and its different constituents to help the user in detecting functionally relevant variants at the protein level.

## 2. Methods

### 2.1. Obtaining 3D Structures of Proteins

The 32 core ADME genes according to PharmaADME [5,25] (Appendix A) and the 23 genes from PharmVar (www.pharmvar.org, accessed on the 23 January 2020) were screened to identify available 3D structures in the Protein Data Bank (PDB). PharmVar aims to centralize the information regarding actionable pharmacogenes by integrating data from the Pharmacogenomic KnowledgeBase database (PharmGKB) and the Clinical Pharmacogenetic Implementation Consortium (CPIC). The criteria of selection among different structures of one protein included resolution, coverage, and completeness. All the structures were stripped from any heteroatoms including ligands, cofactors, and ions. Then, we used MODELLER [26] homology modeling software to predict the 3D coordinates of missing segments and atoms. In addition, ADME core proteins with no experimental 3D structure were processed with homology modeling. Proteins with poor alignment quality between the template and the target are filtered. We generated 20 conformations starting from different random seeds of which we selected the structure with the best DOPE score [27]. Then, stereochemical quality was verified by establishing the Ramachandran plot [28].

### 2.2. Constructing and Evaluating A Predictive Model for Variant Effect Prediction

We compiled a dataset of mutations with experimentally determined ΔΔG values (difference between the folding energy levels of the reference structure and the mutated structure). The dataset is a combination of the Capriotti [29], Khan [30], and Guerios [31] datasets. A total of 2614 data points were collected covering a range of ΔΔG values between −12 and 12.7 kcal/moL and consisting of mutations belonging to 96 unique protein structures. High-impact variants are defined as changes in the amino acid sequence that affect the variation of the Gibbs energy by increasing or decreasing it beyond the threshold value of ±0.5 kcal/moL. We conducted a filtering process to remove outliers based on the calculated ΔΔG and the vibrational entropy difference between the reference structure and the variant ($\Delta \Delta S_{vib}$), which leads to retaining 2015 data points from an initial number of 2614 to use for training and validating the model. For each mutation, we computed 12 features including ΔΔG, $\Delta \Delta S_{vib}$, the solvent-accessible surface area (SASA) ratio between the reference and the variant amino acids, the Position-Specific Scoring Matrix (PSSM) score for substituting the reference amino acid by the variant and by itself, respectively, Sneath’s index, Grantham’s index, and BLOSUM62 substitution score, amino acid volume descriptor (BIGC670101) from AAindex, hydrophobicity descriptor (JOND750101) from AAindex [32], and the difference of the total protein solvent-accessible surface area.

The dataset was split into training (75%) and test (25%) datasets. The training dataset was used to build the model and fit the parameters, while the test dataset was used to evaluate the performance of a trained model and detect overfitting that renders the classifier unusable for prediction purposes. Mutations showing values of −0.5<ΔΔG<0.5 kcal/moL were labeled as neutral, while mutations with ΔΔG outside that range are regarded to cause high-impact changes on the protein function. The predictive model was built using the Python library scikit-learn [33]. We tested several algorithms to classify the variants as neutral or high-impact variants. Metrics of performance were calculated over a 10-fold cross-validation process and were used for additional criteria of optimization to build the predictor. Cross-validation helps to determine the stability of the performance of a model by sampling different portions of the data iteratively.

The random forest algorithm performed better than any other tested approach. Hyperparameter optimization resulted in an adjustment of the number of trees to 1000, a minimum number of samples used to split the nodes to 2, and the minimum samples required to be at a leaf node of 42. The maximum number of features for the best tree split is defined by the square root of the total number of features. The maximum depth of the tree is set to 60, and bootstrap samples were used to build the model.

### 2.3. Evaluation of the Classifier’s Performance Using a Benchmarking Dataset

A list of variants, whose functional effects on the protein are known, were collected as the benchmarking dataset. These include 64 variants of *DPYD* gene characterized by the study of Shrestha et al. [34]. ’High-impact’ and ’neutral’ variants were defined by a threshold activity change of 70% as suggested by the authors in their original paper. We also collected a set of 55 variants belonging to different ADME genes from the CYP P450 superfamily including *CYP2D6*, *CYP2B6*, *CYP2A6*, *CYP2C9*, *CYP2C19*, *CYP2E1*, and *CYP3A4* from the PharmVar database. For these variants, we used the CPIC functional annotation to assign the classes. In addition, we included 39 more variants belonging to *TP53* and collected from 5 different studies [35,36,37,38,39]. Neutral variants were the ones that present ΔΔG values between −0.5 and 0.5 kcal/mol. The other variants were assigned to the ‘high-impact’ class. Each of the benchmarking variants was submitted to the Variant Effect Predictor [40] webserver to predict the functional binary classification using SIFT, PolyPhen-2, PROVEAN, MetaSVM, CADD, and FATHMM. The prediction tools run their default settings within the web server, and their returned binary classification was assigned to the ‘high-impact’ and ‘neutral’ classes.

The performance of the classification was evaluated for each tool as well as the SWAAT classifier by calculating the accuracy, sensitivity (true positive rate or TPR), and specificity (true negative rate or TNR) according to the equations,
$$accuracy=\frac{TP+TN}{TP+TN+FP+FN}$$
$$TPR=\frac{TP}{TP+FN}$$
$$TNR=\frac{TN}{TN+FP}$$
with TN, TP, FN, and FP corresponding to the counts of true negatives, true positives, false negatives, and false positives, respectively.

In addition, we established the Receiver Operating Characteristic (ROC) plot using the scikit-lean [33] Python library to calculate the true positive rates, the false positive rates, and the Area Under the Curve (AUC) from the raw scores of the different variant predictors. For SWAAT, we used the class probability score to perform the same calculation.

### 2.4. Dependencies for Working with SWAAT

SWAAT was built and tested on version 20.10.0 of Nextflow [41]. To run the main workflow, certain requirements should be satisfied. The user must have version 3 of Python with installed modules Pandas, Numpy Matplotlib, Bokeh, and biopython [42]. Several other software must be installed and added to the path, including freesasa [43], FoldX [44], EnCoM [45], and stride [46]. These dependencies are required to run the annotation workflow. PRODRES pipline (https://github.com/ElofssonLab/PRODRES, accessed on the 2 February 2022) is required to calculate the PSSMs of the protein sequences to annotate during the preparation process used by the auxiliary workflow. SWAAT uses pre-mapped coordinates data of the genetic positions with their corresponding amino acid positions. The mapping was performed using the Transvar tool [47] running within the auxiliary workflow. The latter screens the amino acids of the canonical protein reference sequence and extracts the genomic positions of their corresponding DNA codons. We have implemented a series of quality check routines in the Python code (prot2genCoor.py) that runs this process to ensure reliability. SWAAT uses GRCh37 assembly to report on the genomic coordinates.

### 2.5. Overall Description of SWAAT

SWAAT is a workflow tool that aims to annotate missense variants of ADME genes. It uses VCF files as input, extracts the information to map the genetic coordinates to protein coordinates, structurally models the missense protein variants, calculates their biophysical and structural properties, and finally reports all this information to assess the status and the pharmacological relevance of the variant. The information provided by SWAAT considers different scenarios by which the variant can lead to a significant impact on the protein function. First, we considered the effect of the variant on the protein stability by reporting the difference of the folding energies between the wild type and the variant (ΔΔG). For such an end, the FoldX software package [44] is used for its calculation accuracy and efficiency. Moreover, SWAAT integrates many qualitative criteria to help to assess the effect of the variant on protein stability. These involve eleven molecular events that include disulfide breakage, introducing a buried proline, replacing a buried glycine, introducing a buried hydrophilic residue, introducing a buried charged residue, switching the formal charge of a buried residue, changing the secondary structure, replacing a buried charged amino acid, switching the exposure state, replacing an exposed hydrophilic residue with a hydrophobic residue, salt bridge breakage, and the induction of large helical penalty in an α-helix structure caused by a substitution to a glycine or a proline. These criteria are part of the assessment approach used by missense3D [48] that were found to be disease-associated features. SWAAT also identifies the residues that are part of a hotspot patch defined as a cluster of amino acids in the 3D space, distant to each other by at most 6 Å and showing a folding energy difference of at least 2 kcal/moL when substituted to alanine.

Second, a variant can induce a perturbation in the conformational space of a protein which can cause a population shift in the free energy landscape [18]. The perception of biomolecular systems exerting a biological function as rigid entities has been discarded since long ago [49]. Proteins can populate various states at different levels of the energy landscape. This confers the plasticity that is required to undergo complex functional mechanisms such as allostery, domain–domain movements, and structural arrangement to accommodate the ligand in the interaction site. The conformational landscape can be assessed using simulation techniques such as molecular dynamics [50] and Monte Carlo methods [51]. However, these methods are laborious, computationally intensive, and unsuitable for screening variants even for a limited number of genes. To account for the large-scale conformational movements, SWAAT integrates the calculation of the protein normal modes using ENCoM [45]. Normal modes approximate the conformational space to a quadratic potential where the protein oscillates around an equilibrium conformation at low frequency [52]. Unlike other methods, ENCoM accounts for the diversity of amino acids thanks to a specific set of coarse grain parameters, thus allowing to study the effect of mutations. The eigenvectors calculated by ENCoM are used to compute $\Delta \Delta S_{vib}$.

Finally, the user will be able to annotate the genetic variants of ADME genes based on their putative role in binding drugs. We integrated the FTMap [53] data that provide information about the drug interaction hotspots in each of the ADME proteins. These hotspots consist of a set of amino acids evaluated for their capacity to bind 16 probe molecules. SWAAT reports the Z-score and Percentile score statistics as well as the number of hits per hotspot to allow for the quantitative differentiation of residues that are potential drug-binders.

### 2.6. Implementation

SWAAT consists of a set of Python scripts and Bash instructions whose series of execution is managed within Nextflow’s DSL [41]. An auxiliary workflow was developed to build a database that contains information retrieved during the annotation process (Figure 1A) (https://github.com/hothman/SWAAT/tree/master/auxiliary_wf, accessed on the 2 February 2022). A database for the 36 ADME genes was prepared and made available in the main repository of SWAAT (https://github.com/hothman/SWAAT/tree/master/database, accessed on the 2 February 2022). Therefore, users who need to annotate these ADME genes can use the built-in database without running the auxiliary workflow. The auxiliary workflow was made available for versatility purposes in the case where users desire to annotate another set of genes. For example, one of the potential applications is the annotation of a set of functionally related genes implicated in a signaling pathway or a generic biological function. In such a case, the user needs to provide a list of the Uniprot identifiers and the PDB structures corresponding to the genes to annotate. Then, the auxiliary workflow can be run to generate and aggregate the information required by the main workflow for the annotation. These include protein to gene mapping, protein to structure mapping, and other sources of data. One of the limitations of this process is the necessity of large database files required to run the PRODRES pipeline in order to generate the PSSMs. However, the user may skip this process and can provide these files manually by submitting the sequences of the proteins to the webserver version of PRODRES (https://prodres.bioinfo.se, accessed on the 2 February 2022). To get the information about the drug interaction hotspots, the user needs to submit the PDB files to the FTMap server before running the auxiliary workflow [53].

The core functionality of SWAAT includes the main workflow (Figure 1B) that can process a list of variants, annotate them according to sequence-based and biophysical-based properties, and return a detailed report in HTML and CSV format. The user needs only to provide the list of variants in separated VCF files relative to each gene to annotate. The user can restrict the analysis to limited genes by providing a list of their corresponding Uniprot identifiers. The most important parameter used by SWAAT is the path to the database files generated by the auxiliary workflow. These include data about the amino acid sequence, the protein to genome mapping, the PDB to protein sequence mapping, the hotspot patch members, the PSSM files, precalculated BLOSUM62, Grantham, and Sneath scores, and the normal modes of the reference structures.

## 3. Results

### 3.1. Annotated Genes

We have screened 32 core ADME genes defined by the PharmaADME standards as *very important* and involved in drug processing with a high level of evidence. In addition, we also screened all the 23 genes characterized by the PharmVar consortium [54]. The default set of genes annotated by SWAAT comprises 36 members (Figure 2 and Appendix A). They include 2 members of the Arylamine N-acetyltransferase family, 20 CYP P450 members, 3 genes from the Glutathione S-transferases superfamily, and 4 UDP-glycosyltransferase genes. Other members include the *DPYD*, *ABCG2*, *SDNT*, *SULT1A1*, *SCF15M2*, *NUDT15*, *TMT*, and *NUDT15*. The annotated gene list covers 22 of the total 23 genes from PharmVar, as we were not able to obtain a high-quality homology model for CYP4F2. A maximum coverage by the protein structure of 100% was obtained for NAT-1, NAT-2, and CYP2W1. All the CYP P450 proteins show coverage of more than 85%, while the least value was 31%, shown by UDP-glycosyltransferase family members. This was mainly caused by incomplete 3D structures or the availability of only template structures that partially align with the target protein. Partial coverage may also be imposed by the experimental condition during the expression/production assays that disables the inclusion of the full sequence of the protein. For the CYP P450 group, all the proteins lack the transmembrane N-terminal segment. In fact, structures not lacking these segments were excluded because of their significant poor quality compared to other truncated solutions.

Regarding ADME proteins that we modeled using comparative 3D prediction, we obtained high structural similarity between the different sampled conformers ranked according to their DOPE scores. Structures differ only marginally by their backbone atoms, and most of the deviation was reported for rotameric states of the side-chain atoms.

### 3.2. Building and Assessing the Predictive Model

We have implemented a machine learning algorithm operating within SWAAT that uses 12 calculated features to predict if a variant has a high impact on the protein structure (Figure 3A). The predictive model is presented with an accuracy of 73% and a precision score of 0.73. Moreover, we obtained a sensitivity of 0.89 and 0.39, respectively, for the high-impact variant and neutral-variant classes.

Analysis of feature importance showed a high contribution of the ΔΔG, the ratio of the accessible surface area between the variant and the wild-type forms, and $\Delta \Delta S_{vib}$ (Figure 3A). The contribution of features calculated from sequence information also showed an important weight including the data from the PSSMs and the BLOSUM, Sneath, and Grantham scoring indexes. The two physicochemical properties that showed significant importance in increasing the predictive power of the model are the hydrophobicity and volume of the amino acid. These latter features require only the information about the amino acid type for the calculation without the need for the 3D structure.

### 3.3. Benchmarking SWAAT Using Adme and Tp53 Variants

Variant effect prediction tools are widely used to detect high-impact amino acid substitutions. Their wide usage results from the efficiency in screening high numbers of potentially functional variants across the entire human genome. Our tool is better applied for lower throughput applications compared to the variant effect predictors. A comparative report with these tools may provide an insight into the performance of SWAAT. Moreover, training and testing datasets used to build the predictive model of SWAAT include non-ADME proteins and other non-human proteins. Therefore, it is important to evaluate the performance of SWAAT on a dataset that includes ADME genes. To compare the performance of the random forest classifier with the variant effect prediction tools, we included a dataset of variants belonging to *DPYD* (*n* = 64), TP53 (*n* = 39), and others from the CYP P450 superfamily (*n* = 55) (Appendix A). The variants have been annotated with SIFT, PolyPhen-2, CADD, FATHMM, MetaSVM, and PROVEAN. SWAAT’s classifier showed better accuracy of 66% (Figure 3B), which is slightly higher than SIFT (61%), PolyPhen-2 (60%), and PROVEAN (60%). A slight discrepancy was noted between the calculated accuracy and the AUC. SWAAT performed better than any of the sequence-based tools with an AUC of 0.75 except PolyPhen-2 (AUC = 0.76), where it was marginally surpassed (Figure 4, Appendix A).

Overall, SWAAT’s predictive model also outperformed the other tools in terms of sensitivity with a score of 0.84 (Figure 5). With a score of 0.45, the classifier was slightly exceeded by SIFT, PolyPhen-2, and PROVEAN in terms of specificity. Compared to SWAAT, the worst performance in terms of sensitivity was noted for MetaSVM, FATHMM, and CADD (a score of more than 0.5 representing the performance of a random classifier).

In terms of both sensitivity and specificity, SWAAT and PROVEAN had comparable performances on *DPYD* variants. Although it shows slightly poorer sensitivity values compared to MetaSVM, PolyPhen-2, and SIFT, the specificity score of SWAAT’s classifier was better than any other variant prediction tool. For *TP53*, which is a non-ADME gene, SWAAT outperformed all the other tools, showing a better balancing between the sensitivity and the specificity. CADD had a large specificity score of 0.86. However, it performed very poorly in terms of sensitivity, showing a score of less than 0.1. The capability of SWAAT to predict the high-impact variants was noticed for CYP P450 proteins. Our tool outperforms all the other methods with a sensitivity score of 0.86. However, the specificity of SWAAT was poor with a score of 0.32. Nevertheless, all the tools in the upper left quadrant, namely PolyPhen-2, SIFT, and PROVEAN, show a sensitivity of no more than 0.67. It is also clear that SWAAT has better consistency of performance for all three systems mainly in terms of sensitivity with scores of 0.84, 0.82, 0.86, and 0.86, respectively, for the overall comparison, *DPYD*, *TP53*, and CYP P450 genes. As a comparison, PROVEAN’s performance on *DPYD* and *CYP P450* was considerably divergent both in terms of sensitivity and specificity.

## 4. Practical Application

SWAAT requires the user to provide plain text uncompressed VCF files, which are sliced according to the position of the ADME genes in the genome. The annotation uses the GRCh37 genome build, and the user needs to bring the genome position to the same reference before any use. The VCF files must be named according to the gene symbol followed by the file extension (.vcf). For example, the files CYP3A4.vcf and DPYD.vcf are acceptable file names for *CYP3A4* and *DPYD* genes, respectively. The path to the directory that contains the VCF files can be specified via --vcfhome argument.

The path to the database must also be provided with the --dbhome argument. The database is a directory that contains information retreived during the annotation process. The database for the ADME genes is maintained and provided within the GitHub repository of SWAAT (available in the link https://github.com/hothman/SWAAT/tree/master/database, accessed on the 2 February 2022). Notice also that the database can be generated using the auxiliary workflow. The user can also specify the location of the directory that will contain all the output files with the --outfolder option. All these options (i.e., --vcfhome, --dbhome, --outfolder, and others) can be tuned by the user to annotate the default list of ADME genes or to use another personalized list of genes. A typical run of SWAAT annotation can be performed as follows:


  nextflow run main.nf --dbhome /home/hothman/SWAAT/database/ \



      --vcfhome /home/hothman/SWAAT/vcfs \



      --outfolder ./swaat_out \



      --genelist ./inputexample/gene_list.txt.


The output consists of CSV and HTML reports that help the user to explore the collected data for each of the annotated genes. The CSV file offers richer content than the HTML file and serves for elaborate analysis and filtering. The HTML report is better in summarizing the structural events with likely significant impact. These can be spotted as red flag tags in the output (Figure 6A). The report also links the genome coordinates with the amino acid substitution. In addition, the report offers the sequence annotation associated with each amino acid position as well as statistics about the drug-binding likelihood of the amino acid. These statistics were calculated from the FTMap output, which reports the number of probe molecules that likely bind to the amino acid. The HTML report also provides the option to generate interactive plots (Figure 6B). Data files that could help for tracing the workflow’s input/output streams are also reported per gene.

## 5. Discussion

Until very recently, integration of the structural-based large-scale analysis of genetic variants was limited by the availability of data and the efficiency of algorithms. Several initiatives have emerged to fill this gap by providing mapping tools between the genome and the proteome, recommendations for data standardization, updating, and integration [55]. However, these initiatives have not evolved to the development of a complete set of tools allowing the use of the protein structure information in interpreting the functional impact of variants [56]. As far as we know, SWAAT is the first tool to allow this type of analysis using the standard VCF file as input, which is commonly used to store the genetic variability data. In part, this is due to the limited number of genes to annotate (*n* = 36), which makes it relatively trivial to manually model and curate the structures of the ADME proteins. In addition to the sequence-based features provided by SWAAT, the integration of structural properties aimed at capturing molecular events related to low-scale molecular movements such as the side chain rotation and events related to large conformational movements. For the former, the FoldX calculation method offering a good balance between accuracy and efficiency was integrated into the workflow. In addition, ENCoM has been used to calculate the changes in the vibrational entropy upon mutation [57] due to its computational efficiency and because it allows the calculation of normal mode analysis by considering the chemical diversity of the amino acids.

Annotation with SWAAT has several limitations that are worth mentioning. While it is intended for versatility of use, manual protein structure prediction and generating input files for the drug-binding hotspot calculation are still needed. However, the ADME genes within SWAAT are maintained and curated regularly, and the user does not have to use the auxiliary workflow to proceed with the annotation. Moreover, because the predictive machine learning model was trained on a set of single amino acid mutations, SWAAT cannot annotate scenarios of proteins with multiple variants or deletions. In addition, the range of variants within the genes to annotate by SWAAT depends on the coverage by the protein structure. For example, our workflow is capable of annotating all the coding variants of *NAT-1* and *NAT-2* genes while only 28% of *SLC15A2* variants are available for annotation due to the low coverage by the corresponding protein structure. Nevertheless, all the non-processed variants are reported in the output of SWAAT. The current version of SWAAT does not handle the annotation of multiple isoforms encoded by a single gene nor does it account for the availability of discontinuous structures of a single protein. For example, structures of two domains from the same protein can be solved separately using experimental methods. One of the workaround solutions is to provide separate structures for the annotation workflow for as many runs as needed to annotate the corresponding variants. In upcoming versions of SWAAT, we will implement the option to use multiple structures per gene.

The integration of sequence-based and structure-based features in predictive machine learning models has a significant impact on the performance of the model. In particular, the calculation of the normal modes has increased the predictive power. The importance of vibrational entropy might be protein-dependent and might not be relevant in the case of highly packed proteins [58], which leads some studies to discard it from the calculation due to computational cost [59]. However, in our case, it emerges as a feature with a significant impact on the predictive power of the model but not as important as other features such as the calculation of the accessible surface area surface. Since ENCoM is reasonably efficient, especially for medium and small size proteins, the integration of the vibrational entropy was not a limitation in the annotation process.

None of the sequence-based annotation tools use the structure-based features described in Figure 3A. In addition, chemical and physical descriptors are exclusively used by SWAAT in its annotation process. However, there are some similarities with the sequence-based features. For example, SIFT exploits the information of amino acid conservation of the PSSMs to calculate a probability score per position [60]. In addition, PolyPhen-2 integrates similar types of information to predict the impact of non-synonymous SNPs [61]. PROVEAN integrates information about amino acid conservation from substitution matrices to calculate a delta score that measures the effect of the variation. CADD, on the other hand, uses a support vector machine predictor trained over millions of human variants to calculate a score that assesses deleteriousness [62]. The effect of amino acid substitution is calculated from evolutionary information using Hidden Markov Models with FATHMM [63], and MetaSVM is unique in its concept as it performs a meta-analysis from multiple OMICS data to calculate a prediction score for the variant’s impact [64].

Conventional variant effect predictors perform poorly in annotating ADME genes that are generally highly variable in population groups and are not always disease-related [65]. This makes extracting conservation patterns difficult. However, most of the variant impact prediction tools are designed to detect pathological impact based on evolutionary constraints. In such regard, even though SWAAT uses conservation information, structure-based features are still the major determinants of the random forest classifier’s performance, which makes it more suitable for annotating ADME genes than any other conventional tool and could explain its remarkable consistency to annotate different genes.

In general, the SWAAT predictor performed better than other sequence-based approaches in terms of sensitivity, specificity, and consistency. We have obtained a similar value of sensitivity for the *DPYD* gene evaluated by sequence-based methods obtained from the analysis by Shrestha et al. [34], thus showing the reproducibility of our results. However, SWAAT has a lower accuracy score of 67% compared to their variant predictor tool, DPYD-Varifier (85%). This is expected, since their training dataset is exclusive for DPYD and includes functional biochemical characterization related to 5-fluorouracil toxicity, whereas SWAAT-predictor training data are generic. However, we were able to obtain better sensitivity for SWAAT (0.82) compared to DPYD-Varifier (0.73). Our work highlights the potential of using structural-based methods in predicting the effect of the variants. The ability of SWAAT to discriminate true positive variants with functional effect on the CYP P450 superfamily is a significant achievement. CYP P450 proteins include, arguably, the most functionally relevant ADME genes (91% of the total genes from the PharmVar database) involved in the drug metabolization process. This could be an advantage for variant prioritization in clinical applications, as more sequencing will reveal new polymorphisms in many populations.

The coverage of 95% of the PharmVar database expands the clinical applicability of SWAAT. Efforts within the consortium have been focused on providing standardized nomenclature of clinically actionable genes for which the plans have been established by the Clinical Pharmacogenetics Implementation Consortium and the expert panel from PharmGKB. The expansion of high-throughput sequencing will lead to the identification of more novel variants and variants with unknown significance belonging to actionable genes. In this regard, our tool draws a new perspective for uncovering variants with pharmacological impact, leading to better accuracy and specificity of the annotation process and an increase in the predictive power.

With the continuous enrichment of the Protein Data Bank, the improvement of experimental protein-solving approaches, and the significant leap toward solving the protein-folding problem, integration of structural analysis in variant impact prediction will become easier to achieve. The entire human proteome would be available in the upcoming future, and new tools to integrating structural analysis within genome analysis workflows are necessary. Based on this, SWAAT has been created as a part of benchwork tools developed by our team [66] to refine the annotation of ADME gene variants, thus improving the decision-making process in precision medicine practices.

## 6. Conclusions

We described SWAAT, a tool to annotate missense coding variants from ADME genes based on structural features. Comprehensive and detailed reports are generated by the workflow to present the user with reliable information concerning the functional impact of the annotated variants. SWAAT integrates a random forest predictive model that showed good performance compared to other sequence-based variant prediction tools. The auxiliary workflow could be employed to customize the annotation of other genes outside the ADME list. SWAAT could be applied to the discovery, prioritization, and fundamental understanding of the impact of putative actionable variants with clinical interest. In the upcoming future, we are willing to address many of the limitations of SWAAT and develop a webserver version for the tool. 

## Figures and Tables

**Figure 1 jpm-12-00263-f001:**
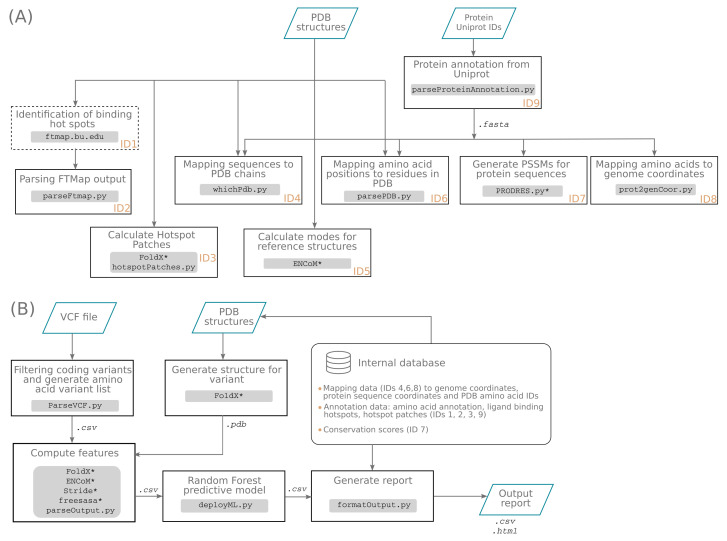
Description of the SWAAT toolset. SWAAT consists of an auxiliary workflow (**A**) that helps to prepare an internal database and the main workflow (**B**) that annotates the variants and predicts their effect on the protein structure. The processes in the auxiliary workflow are assigned to IDs that describe their functionalities in the internal database. “Identification of binding hotspots“ is an external process that runs at the FTMap server. Processes marked by asterisk correspond to external program calls (FoldX, ENCoM, Stride, freesasa and, PRODRES).

**Figure 2 jpm-12-00263-f002:**
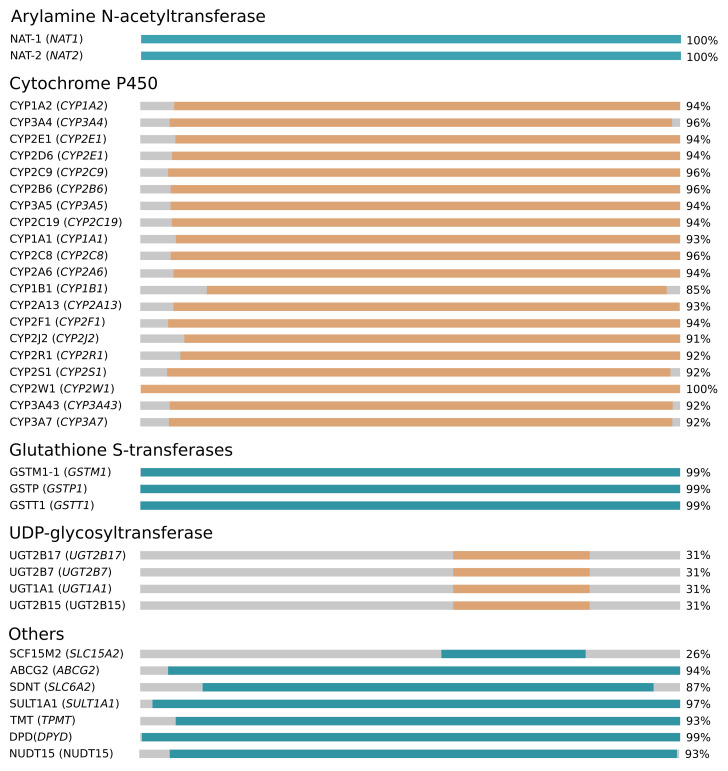
SWAAT’s default annotation ADME gene list. Protein names and gene names are given in the left-most column. We indicate the coverage by the structure. The genes belong to PharmaADME and PharmVar lists. They were screened for a reliable 3D structure of their corresponding proteins. The structures were obtained from the Protein Data Bank or were constructed using homology modeling. Blue and orange colors in the figure do not indicate any information. They are used to allow distinguishing the ADME protein group. Gray areas represent the uncovered protein segments by the 3D structure.

**Figure 3 jpm-12-00263-f003:**
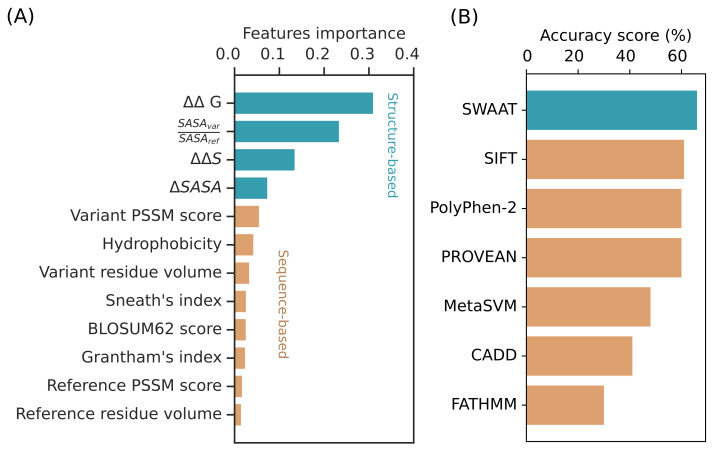
Characteristics of SWAAT’s binary classifier to predict the effect of missense variants. (**A**) Feature importance profile of the random forest classifier. (**B**) Accuracy scores of SWAAT and six other sequence-based variant effect prediction tools performed on a dataset that includes different ADME genes and the *TP53* gene.

**Figure 4 jpm-12-00263-f004:**
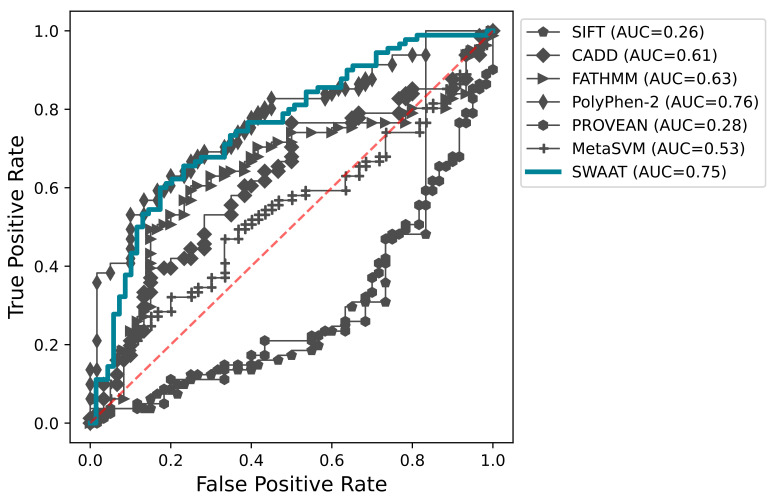
The ROC curve was generated for each tool, and the AUC is shown in the figure’s legend between parentheses. Note that the calculation was performed for the entirety of the benchmarking dataset. The red discontinuous line indicates the performance of the random classifier.

**Figure 5 jpm-12-00263-f005:**
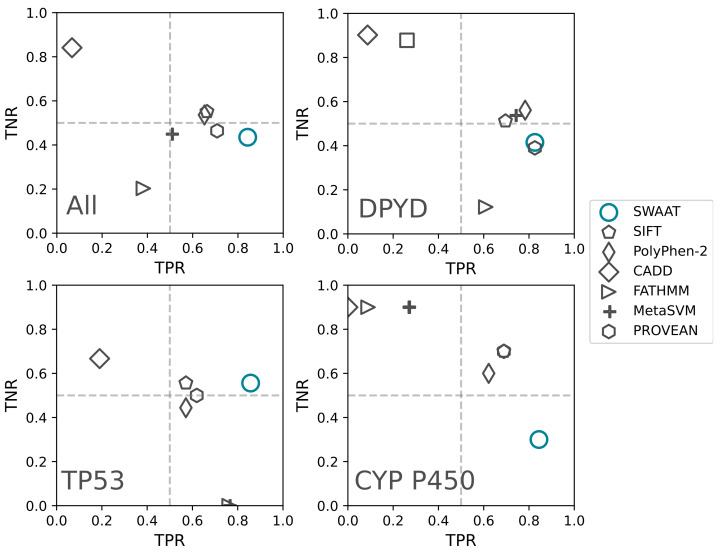
Sensitivity vs. specificity plots of the random forest predictive model integrated within SWAAT in comparison to SIFT, PolyPhen-2, CADD, FATHMM, MetaSVM, and PROVEAN. Variants from *DPYD*, *CYP2D6*, *CYP2B6*, *CYP2A6*, *CYP2C9*, *CYP2C19*, *CYP2E1*, and *CYP3A4* are included in the analysis. In addition, variants of *TP53* were used to evaluate the performance of SWAAT’s classifier based on well-established thermodynamics data of the gene mutants. The vertical and the horizontal discontinuous lines indicate the domain of values expected to be shown by a random classifier.

**Figure 6 jpm-12-00263-f006:**
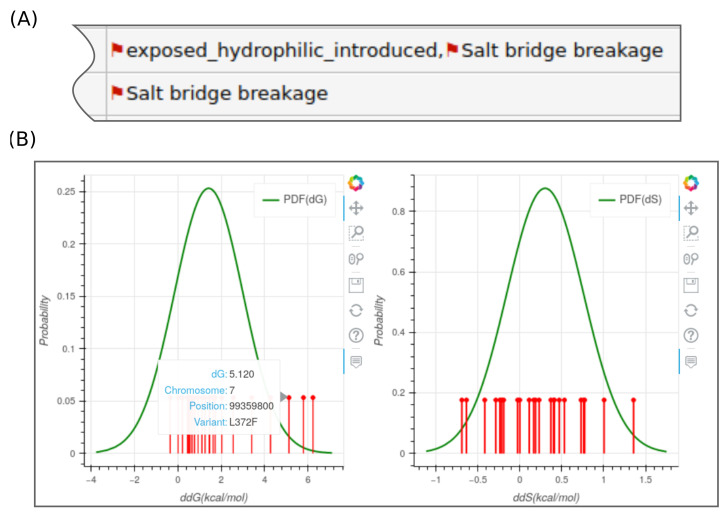
HTML report of SWAAT. (**A**) Rows in this figure correspond to variants. Red flags are tags that show structural events that can result in a deleterious effect on the protein structure. (**B**) Interactive plots generated by SWAAT. Lolliplots correspond to variants located at different levels of ΔΔG and $\Delta \Delta S_{vib}$. The user can hover over these elements to display useful information. The distributions were calculated by fitting a Gaussian model to a set of computed estimations from mutated forms of different proteins.

## Data Availability

The code of the annotation workflow and the auxiliary workflow are available from the GitHub repository https://github.com/hothman/SWAAT, accessed on 1 January 2022. We also provided detailed instructions and tutorials in the same repository.

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
