# Peer review of "SWAAT Bioinformatics Workflow for Protein Structure-Based Annotation of ADME Gene Variants"

_jpm, 2022, doi:10.3390/jpm12020263_

Round 1
Reviewer 1 Report
The authors developed a bioinformatics workflow for protein structure-based annotation of ADME gene variants. It is an impressive work and takes a lot efforts. Here are a few comments and questions I have for the paper.
- Line 79, "We generated 20 conformations starting from different random seeds of which we select the structure with the best DOPE score". If a structure is very stable, this structure and very similar structures should appear several times in the 20 conformations. The results would be more likely to be reproduced in future. Does the structure with the best DOPE score appear multiple times in the 20 conformations? If not, this method may not be very stable or 20 times is not enough.
- The authors considered the effects of the variant on protein. One gene may produce multiple proteins. Does the authors select one protein associated with the variant or calculate an average effect of all proteins?
- Line 100, "We tested several algorithms for their performance to classify the variants as neutral or high-impact variants and we used a 10-fold cross-validation process to evaluate the performance of the model on the test dataset." The authors have divided the data into training (75%) and testing (25%) sets. We usually use cross validation in training dataset to tune the parameters in the model and then build the model with the tuned parameters in the training dataset. The performance of this model is estimated in the testing dataset. I am not sure how 10-CV were used in testing dataset. Please give more details.
- Figure 1, what is the difference between blue and yellow areas?
- Figure 4, although SWAAT has hight sensitivity, the specificity is relatively low, especially in DPYD and CYP P450. It is hard to compare these methods in figure 4. I am not familiar with all these methods, can we calculate an AUC from them? What is the accuracy of these methods?
- I did not find any supplementary materials. I am not sure whether the authors did not upload the supplementary or the journal did not share them with me.
Author Response
1. Line 79, "We generated 20 conformations starting from different random seeds of which we select the structure with the best DOPE score". If a structure is very stable, this structure and very similar structures should appear several times in the 20 conformations. The results would be more likely to be reproduced in the future. Does the structure with the best DOPE score appear multiple times in the 20 conformations? If not, this method may not be very stable or 20 times is not enough.
Indeed, in all the instances we obtained a great structural similarity between the different sampled conformers ranked according to their DOPE scores. Structures differ only minorly regarding their backbone atoms while they diverge mainly with their rotameric side-chain atoms. One thing to pin-point is that the optimization process with MODELLER does not result in divergent sampled structures due to the enormous number of spatial restraints used in model building.
We have added this to the text in lines 283-287
2. The authors considered the effects of the variant on protein. One gene may produce multiple proteins. Does the authors select one protein associated with the variant or calculate an average effect of all proteins?
We thank the reviewer for this comment. For each ADME gene that we have modeled in the default list, we only considered the longest protein isoform in terms of sequence coverage. As for overlapping genes, they can be handled from the input since the user can choose the genes set to annotate by assigning the input VCF files to their gene symbols. The default gene list has no internal overlap per se. However, we admit that SWAAT should handle multiple proteins per gene. Proteins annotated by SWAAT must have continuous structures. Some proteins can be only crystallized by their discontinuous domains. In another case, one might be interested in different conformations of the same protein. We have pointed out these limitations in the last paragraph of the discussion lines 409-416. Therese issues will be tackled in future versions of the workflow.
3. Line 100, "We tested several algorithms for their performance to classify the variants as neutral or high-impact variants and we used a 10-fold cross-validation process to evaluate the performance of the model on the test dataset." The authors have divided the data into training (75%) and testing (25%) sets. We usually use cross-validation in training dataset to tune the parameters in the model and then build the model with the tuned parameters in the training dataset. The performance of this model is estimated in the testing dataset. I am not sure how 10-CV was used in the testing dataset. Please give more details.
Thank you for underlining this point. The cross-validation was indeed used to tune the parameters of the model and compare different types of predictions in the building phase. The statement related to the cross validation was mentioned to highligh that the different scores used for the evaluation of the model’s performance were obtained from a cross-validation process rather than a single point measuring. We now refer to the cross-validation as suggested by the reviewer in lines 125-128.
4. Figure 1, what is the difference between blue and yellow areas?
Colors in the figure do not indicate any information. They are used to allow for easy visual separation between the ADME protein group.
We have indicated this information in the label of figure 2 (Figure 1 in the older version).
5. Figure 4, although SWAAT has high sensitivity, the specificity is relatively low, especially in DPYD and CYP P450. It is hard to compare these methods in figure 4. I am not familiar with all these methods, can we calculate an AUC from them? What is the accuracy of these methods?
The accuracy score is now mathematically described in the Methods section (lines 155-159). We have now added a new figure (figure 4) to describe the performance of SWAAT’s classifier in terms of ROC curve and AUC. We established this calculation based on the raw scores of the sequence-based variant prediction tools and the class probability score of SWAAT. Overall, we found a good agreement with the accuracy scores calculated in Figure 3, except that in terms of AUC, PolyPhen-2 very slightly outperformed SWAAT. We have also uploaded the raw data and the Python code of this analysis to enable reproducibility (Supplementary data 3 and supplementary material 4). We have described these results in lines 317-321.
6. I did not find any supplementary materials. I am not sure whether the authors did not upload the supplementary or the journal did not share them with me.
We apologize for this error. Apparently, the supplementary files were not uploaded in the last version. We uploaded them in the current version.
Reviewer 2 Report
Authors present the workflow for annotating Absorption, Distribution, Metabolism, and Excretion (ADME) gene targets to further predict the variant effect based on structural properties. The coding tools were developed using Python stored in GitHub. Results showed that 0.73 of accuracy was performed on a test dataset by using a random forest binary classifier of a machine learning predictive model. This paper is interesting and might be useful for drug delivery and targeting in the future. There are comments to improve the study that should be addressed.
1 Introduction must be improved, as follows:
First line of introduction at “ADME (Absorption, Distribution, Metabolism, and Excretion)” for English Abbreviation could be changed into “Absorption, Distribution, Metabolism, and Excretion (ADME)”.
Introduction is missing the details of a machine learning (ML) predictive model, and why ML was chosen and used for classifying the test dataset, and what parameters or annotating targets were classified and why they are important to the future of the drug targeting and delivery.
2 How many classifiers were used in this study?
3 How to calculate 0.73 accuracy performed by a random forest binary classifier? Presents an accuracy in term of percentage might be better off, e.g., 73% of accuracy. However, the scale in Figure 4 is "0-1".
4 What are the 6 commonly names of "SWAAT outperformed 6 commonly used sequence-based variant prediction tools in 17 terms of sensitivity and has comparable specificity”? How to calculate sensitivity and specificity?
5 the site of URL http://www.pharmaadme.org could not be accessed, please make sure the right URL, is it http://pharmaadme.org instead of? However, both URLs could not be accessed. As it is important to reference the ADME standard.
6 Results at 3.1 Annotated genes is well-designed. However, was Figure 1 obtained from analysis? Or was it obtained from the reference of the ADME standard?
7 Results at 3.2. Implementation should not include in the result section and also Figure 2.
8 3.3 Building and assessing the predictive model, and 3.4 Benchmarking SWAAT for ADME genes and TP53 are mixed the analysis progresses, e.g., ML details should be written in methodology section.
9 Have authors excused all these coding: SIFT, PolyPhen-2, CADD, FATHMM, MetaSVM, and PROVEAN? Where do they come from? They are missing information in the methodology section.
10 At Figure 4 Caption, have authors developed “the SWAAT classifier in terms of sensitivity and specificity”? or Have authors developed the SWAAT tools as a classifier for annotating Absorption, Distribution, Metabolism, and Excretion (ADME) gene targets?
11 Please check the abbreviations at lines 428-431.
12 Conclusion section should be presented all significant results.
Author Response
1 Introduction must be improved, as follows:
First line of introduction at “ADME (Absorption, Distribution, Metabolism, and Excretion)” for English Abbreviation could be changed into “Absorption, Distribution, Metabolism, and Excretion (ADME)”.
This is now modified in the revised version.
Introduction is missing the details of a machine learning (ML) predictive model, and why ML was chosen and used for classifying the test dataset, and what parameters or annotating targets were classified and why they are important to the future of the drug targeting and delivery.
We have included a new paragraph in the introduction at lines 66-78 to cover all these aspects.
How many classifiers were used in this study?
In this study we used 7 classifiers including the one integrated within the annotation workflow that is subject to our discussion in the paper. The other classifiers consist of widely used variant effect prediction tools in genomics, namely SIFT, PolyPhen-2, CADD, FATHMM, PROVEAN and MetaSVM. These however are not part of SWAAT, but were used to compare our classifier to their performance to put some perspectives about its utility. We have included a new description in the ‘Methods’ section to explain in detail this important step in lines 140-164.
How to calculate 0.73 accuracy performed by a random forest binary classifier? Presents an accuracy in term of percentage might be better off, e.g., 73% of accuracy. However, the scale in Figure 4 is "0-1".
The equation that allowed the calculation of the accuracy is now presented in the Methods section (After line 157). We have changed figure 3 to show the accuracy scores as percentages.
What are the 6 commonly names of "SWAAT outperformed 6 commonly used sequence-based variant prediction tools in 17 terms of sensitivity and has comparable specificity”? How to calculate sensitivity and specificity?
The list of these tools is now added to the abstract. The calculation of the sensitivity and specificity are now detailed in lines 155-159.
the site of URL http://www.pharmaadme.org could not be accessed, please make sure the right URL, is it http://pharmaadme.org instead of? However, both URLs could not be accessed. As it is important to reference the ADME standard.
As it appears, by the time the paper has been processed, the link to the pharmaadme gene list has become obsolete. We, therefore, refer to the papers by Hovelsen et al (2017) and da Rocha (2021) to reference the list. The list was also uploaded in supplementary data 1.
Results at 3.1 Annotated genes is well-designed. However, was Figure 1 obtained from analysis? Or was it obtained from the reference of the ADME standard?
The list of genes in figure 1 contains a summary of all the genes that are annotated by default with SWAAT. These were obtained by screening the core genes from the PharmaADME list and PharmVar consortium to search for a representative and accurate 3D structure of their corresponding proteins. We have added all these details to the label of figure 2 (Figure 1 in the previous version).
Results at 3.2. Implementation should not include in the result section and also Figure 2.
The ‘Implementation’ section and figure 1 (Figure 2 in the previous version) were moved to ‘Methods’ in lines 228-261
3.3 Building and assessing the predictive model, and 3.4 Benchmarking SWAAT for ADME genes and TP53 are mixed the analysis progresses, e.g., ML details should be written in methodology section.
Much of these sections now are integrated into the methods (lines 107-112 and lines 182-227) to describe the building of the model according to the suggestions by the reviewer.
Have authors excused all these coding: SIFT, PolyPhen-2, CADD, FATHMM, MetaSVM, and PROVEAN? Where do they come from? They are missing information in the methodology section.
We now dedicate a full paragraph in methodology to describe this point and how these tools may be relevant to our study in lines 140-154.
At Figure 4 Caption, have authors developed “the SWAAT classifier in terms of sensitivity and specificity”? or Have authors developed the SWAAT tools as a classifier for annotating Absorption, Distribution, Metabolism, and Excretion (ADME) gene targets?
The SWAAT classifier is only one part of SWAAT. It compiles some of the parameters compiled by the workflow and processes them using a random forest machine-learning algorithm to predict the class of the variant (High impact or deleterious). SWAAT also includes other modules to calculate and assign functional properties for the variant such as the variation of the folding energy, the variation of the vibrational entropy, and the red flags tags. The PDB structures are integrated into the workflow and used as a primary source to compute all of the properties belonging to ADME proteins. In figure 5 (Figure 4 in the previous version), the performance of the machine learning classifier was evaluated to calculate the specificity and the sensitivity and compare it to 6 other commonly used variant prediction tools. Through in inclusion of TP53 gene, we wanted to assess SWAAT based on available thermodynamics data. We also wanted to show that SWAAT can be used to annotate other genes if the user wishes to. We have changed the form of the description of the figure label.
Please check the abbreviations at lines 428-431.
Thank you for spotting the errors. We have fixed this in the revised version.
Conclusion section should be presented all significant results.
The conclusion has been modified and shortened to contain only the significant outcomes resulting from this study (lines 483-492)
Reviewer 3 Report
1- Our tool also includes a machine learning predictive model based on a random forest binary classifier with 0.73 accuracies performed on the test dataset. Why such a specific classifier (i.e., 0.73)?
2- In figure 1, please define what structural and functional aspects of the proteins indicated have not been covered? Also as to why these particular aspects have not been covered?
3- The author could comment on how the genomic variant for instance in a VCF file information could directly get linked to protein/ aa variants and vice versa? For instance, how confident is an algorithm in predicting the degenerate sequence from the protein sequence? The authors could outline this since in the conclusion they do state that this tool relates 3D protein analysis with genetic variants.
4- In Figure 3, how do the 6 tools vary with respect to the features outlined in A?
5- In figure 4, what is the reason that different tools function so differently based on the gene assayed?
6- In figure 5, ultimately the greatest question to answer about a new method or tool is how accurate and close to reality is its predictions? For instance, the authors could verify the effect of the variants predicted by the tool.
7- In the command line in line 301, please define what --dbhome is located, is this a series of files that the author has compiled that are readily available? I can see that the author state that this is a directory that information applied to accelerate the annotations process. Please state if the readers can modify this, add or omit based on their individual needs? The same applies to --genelist and --vcf home, can these be modified to fit the specific needs of the study?
8- More validation of the tool by corroborating variants predicted with actual effects in genes/ proteins/ cells is required.
Author Response
1- Our tool also includes a machine learning predictive model based on a random forest binary classifier with 0.73 accuracies performed on the test dataset. Why such a specific classifier (i.e., 0.73)?
The random forest classifier was selected after testing also the following algorithms: logistic regression, Support Vector Machine, Decision tree, Naive Bayes, neural network, XGboost, and KNeighbors. For each one of these algorithms, we performed feature engineering and parameter optimization. The random forest classifier showed the best accuracy of 73% compared to the other algorithms.
The referred sentence in the abstract now becomes :
Our tool also includes a machine learning random forest binary classifier that showed an accuracy of 73%.
2- In figure 1, please define what structural and functional aspects of the proteins indicated have not been covered? Also as to why these particular aspects have not been covered?
This was mainly caused by the incomplete resolution of the protein's 3D structure or the availability of template structures that partially align with the target protein. Partial coverages may also be imposed by the experimental condition during the expression/production assays that disable the inclusion of the full sequence of the protein. For the CYP P450 group, all the proteins lack the transmembrane N-terminal segment. We excluded such types of structures because of their significantly poor quality compared to other truncated solutions, which in turn might negatively impact calculations that require the 3D coordinates to run.
We have included this explanation in lines 276-282.
3- The author could comment on how the genomic variant for instance in a VCF file information could directly get linked to protein/ aa variants and vice versa? For instance, how confident is an algorithm in predicting the degenerate sequence from the protein sequence? The authors could outline this since in the conclusion they do state that this tool relates 3D protein analysis with genetic variants.
SWAAT uses pre-mapped coordinates data of the genetic position with their corresponding amino acid position. The mapping was performed using the Transvar tool that runs within the auxiliary workflow. The latter screens the amino acids of the canonical protein reference sequence and extracts the genomic positions of their corresponding DNA codons. We have implemented a series of quality check routines in the Python code that runs this process to ensure reliability.
Section 2.3 was modified to highlight this in a better way (lines 175-180)
4- In Figure 3, how do the 6 tools vary with respect to the features outlined in A?
None of the sequence-based annotation tools use the structure-based features described in Figure 3A. Also, chemical and physical descriptors are exclusively used by SWAAT in its annotation process. There are however some similarities with the sequence-based features. For example, SIFT exploits the information of amino acid conservation of the PSSMs to calculate a probability score per position. Also, PolyPhen-2 integrates similar types of information to predict the impact of non-synonymous SNPs. PROVEAN integrates information about amino acid conservation from substitution matrices to calculate a delta score that measures the effect of the variation. CADD on the other hand uses a support vector machine predictor trained over millions of human variants to calculate a score that measures deleteriousness. The effect of amino acid substitution is calculated from evolutionary information using Hidden Markov Models with FATHMM. MetaSVM is unique in its concept as it performs a meta-analysis from multiple OMICS data to calculate a prediction score for the variant’s impact.
This has been highlighted in lines 427-439
5- In figure 4, what is the reason that different tools function so differently based on the gene assayed?
Conventional variant effect predictors perform poorly in annotating ADME genes that are generally highly variable in population groups and are not always disease-related. This makes extracting conservation patterns difficult. Most of the variant impact prediction tools, however, are designed to detect pathological deleteriousness based on evolutionary constraint type of information. In such regard, even though SWAAT uses conservation information, structure-based features are still the major determinants of the random forest classifier’s performance which makes it more suitable for annotating ADME genes than any other conventional tool.
This is now highlighted in the discussion section (lines 440-448)
6- In figure 5, ultimately the greatest question to answer about a new method or tool is how accurate and close to reality is its predictions? For instance, the authors could verify the effect of the variants predicted by the tool.
We agree on this point. Functional assays, involving testing isolated and purified ADME proteins are perhaps the most reliable source of information that can confirm the impact of genetic variants. However, this is out of our reach in the current work. Nevertheless, more high-quality data could be made available in the future which may be used to improve the predictive model or in benchmarking the performance of SWAAT. The values in figure 5, while theoretical, are not deviating too much from experimental measures of the variation in the folding energy. While not intended as absolute values, they can however serve in discriminating between the false positive and true positive type of variants. Red flag tagging can improve significantly the filtering process.
7- In the command line in line 301, please define what --dbhome is located, is this a series of files that the author has compiled that are readily available? I can see that the author state that this is a directory that information applied to accelerate the annotations process. Please state if the readers can modify this, add or omit based on their individual needs? The same applies to --genelist and --vcf home, can these be modified to fit the specific needs of the study?
We have now added the link to the database home for ADME genes (line 355). We also added the following statement “All these options (i.e. --vcfhome, --dbhome, --outfolder and others) can be tuned by the user to annotate the default list of ADME genes or to use another personalized list of genes.” (lines lines 359-361)
8- More validation of the tool by corroborating variants predicted with actual effects in genes/ proteins/ cells is required.
As we stated also in point (5), experimental validation is very important to provide golden standards for evaluating novel approaches of annotating ADME genes. This however is difficult to achieve within the timeline of this revision to process hundreds of ADME variants.
Round 2
Reviewer 1 Report
The authors have addressed all my comments. The paper should be ready for publication now.
Reviewer 3 Report
The authors have addressed my comments